# Genomic Regions Associated with the Control of Flowering Time in Durum Wheat

**DOI:** 10.3390/plants9121628

**Published:** 2020-11-24

**Authors:** Priyanka Gupta, Hafssa Kabbaj, Khaoula El Hassouni, Marco Maccaferri, Miguel Sanchez-Garcia, Roberto Tuberosa, Filippo Maria Bassi

**Affiliations:** 1Department of Agricultural and Food Sciences, University of Bologna, Viale G Fanin 44, 40127 Bologna, Italy; vidhiguptaniwari@gmail.com (P.G.); marco.maccaferri@unibo.it (M.M.); roberto.tuberosa@unibo.it (R.T.); 2International Center for Agricultural Research in the Dry Areas (ICARDA), 10000 Rabat, Morocco; H.Kabbaj@cgiar.org (H.K.); khaoula.elhassouni@uni-hohenheim.de (K.E.H.); M.Sanchez-Garcia@cgiar.org (M.S.-G.)

**Keywords:** durum wheat, flowering time, pheno-environments, GWAS, QTL

## Abstract

Flowering time is a critical stage for crop development as it regulates the ability of plants to adapt to an environment. To understand the genetic control of flowering time, a genome-wide association study (GWAS) was conducted to identify the genomic regions associated with the control of this trait in durum wheat (*Triticum durum* Desf.). A total of 96 landraces and 288 modern lines were evaluated for days to heading, growing degree days, and accumulated day length at flowering across 13 environments spread across Morocco, Lebanon, Mauritania, and Senegal. These environments were grouped into four pheno-environments based on temperature, day length, and other climatic variables. Genotyping with a 35K Axiom array generated 7652 polymorphic single nucleotide polymorphisms (SNPs) in addition to 3 KASP markers associated with known flowering genes. In total, 32 significant QTLs were identified in both landraces and modern lines. Some QTLs had a strong association with already known regulatory photoperiod genes, *Ppd-A* and *Ppd-B*, and vernalization genes *Vrn-A1* and *VrnA7*. However, these loci explained only 5% to 20% of variance for days to heading. Seven QTLs overlapped between the two germplasm groups in which *Q.ICD.Eps-03* and *Q.ICD.Vrn-15* consistently affected flowering time in all the pheno-environments, while *Q.ICD.Eps-09* and *Q.ICD.Ppd-10* were significant only in two pheno-environments and the combined analysis across all environments. These results help clarify the genetic mechanism controlling flowering time in durum wheat and show some clear distinctions to what is known for common wheat (*Triticum aestivum* L.).

## 1. Introduction

Durum wheat (*Triticum durum* Desf.) is an allotetraploid with A and B genomes. Its major production base is the European Union, with 9 million tonnes harvested in 2018, followed by Canada, Turkey, the United States, Algeria, Mexico, Kazakhstan, Syria, and India [1]. Flowering induction plays a pivotal role in the plant life cycle, affecting reproductive success and yield depending on the prevailing climatic conditions of the target environment. In durum wheat, the heading and flowering times are critical stages in crop development as they play an important role in adaptation, yield potential, and grain quality [2]. In addition, climatic stress during anthesis negatively affects pollen fertility [3,4]. Therefore, plant breeders need effective tools to predict flowering time in order to identify promising genotypes adapted to different environmental conditions.

Flowering time in wheat is controlled mainly by three groups of loci, two of which interact with environmental factors—namely, photoperiod sensitivity genes (*Ppd*) and vernalization genes (*Vrn*) [5]—while the third group of loci is defined as “narrow-sense earliness” or “earliness per se” (*Eps*), because these act independently of vernalization and photoperiod [6]. However, it is not certain whether *Eps* genes act independently of all environmental cues [7,8], since for instance Appendino and Slafer [9] showed that *Eps* genes could respond to temperature changes. Allelic variation in *Ppd* genes divides the temperate cereals into photoperiod-sensitive and photoperiod-insensitive classes, whereas differences in the *Vrn* alleles divide them into winter and spring classes.

Vernalization is the acquisition or acceleration of a plant’s ability to flower after exposure to a certain degree of cold temperature, and it is a strategic adaptation mechanism to postpone heading after the frost-prone winter months [10]. The natural variation in vernalization requirement is mainly associated in common wheat with allelic differences in the *Vrn1*, *Vrn2*, and *Vrn3* genes on A, B, and D chromosomes. The *Vrn* genes regulate the transition from vegetative to reproductive phase in response to temperature [5] and thus determine the spring and winter growth habit. Different alleles respond differently to temperatures, but in general prolonged exposures to temperatures at least below 16 °C are necessary to achieve the vernalization needs, and colder temperatures tend to accelerate this process, reducing the total number of days needed for flowering. This is one of the main adaptation systems that allows winter wheat to survive at lower temperatures than spring wheat [11]. Winter wheat possesses recessive alleles at the *Vrn-A1, Vrn-B1, Vrn-D1*, and *Vrn-D5* loci [12,13], while spring wheat has dominant alleles at one or more of them [14]. The dominant allele of *Vrn-A1* confers complete insensitivity to vernalization in spring growth habit and is epistatic to the dominant alleles of *Vrn-B1*, *Vrn-D1*, and *Vrn-D5*, which confer low sensitivity to vernalization in a facultative winter growth habit [12,15,16,17]. Durum wheat also harbors *Vrn-A1* and *Vrn-B1*, located on the long arms of chromosomes 5A and 5B [18,19]. Recent advances in wheat genomics have allowed for the cloning of the *Vrn-A1, Vrn-B1*, and *Vrn-D1* genes [20] and the development of functional SNP markers for their characterization.

Photoperiod response is another important factor influencing the initiation and length of the flowering period. The natural variation in response to photoperiod is mainly determined by allelic differences in the *Ppd1* gene, a member of the pseudo-response regulator (PRR) gene family [21]. Similarly to vernalization, this a major adaptation mechanism to delay flowering until after the short days of winter have passed to avoid the risk of frost damage to the reproductive organs. Photoperiod-sensitive wheat plants initiate flowering only after long days with more than 13 h of sunlight are perceived, while photoperiod-insensitive types can induce heading irrespective of daylength. Mutation at the *Ppd-1* locus enables wheat to become photoperiod insensitive and flower irrespective of day length. These mutations have been put under strong selection pressure in the past to enhance yield under certain climatic conditions via the promotion of early flowering to avoid terminal climatic stresses late in the season. Particularly famous is the case of Norman Borlaug, who was able to impose a very stringent selection of photoperiod-insensitive types via the use of shuttle breeding (two breeding cycles per year). This process resulted in widely adapted wheat cultivars that prompted the Green Revolution [22,23].

In durum wheat, photoperiod sensitivity is determined by the *Ppd-A1* and *Ppd-B1* loci, located on chromosomes 2AS and 2BS, respectively [24], while photoperiod insensitivity results from mutations in any of the two *Ppd-1* genes. By convention, alleles conferring photoperiod insensitivity are assigned an “a” suffix (e.g., *Ppd-A1a*) [25]. In durum wheat, two large deletions within the *Ppd-A1* gene designated as alleles “GS-100” and “GS-105” were reported to accelerate flowering, which led Wilhelm et al. [26] to conclude that these deletions are the likely causal basis of photoperiod insensitivity in tetraploid wheat.

While the knowledge of gene action to control flowering time in hexaploid wheat is quite accurate and routinely exploited in marker-assisted breeding [27], its application in durum wheat has not yet been fully tested, and several discrepancies such as the actual effect of *Vrn* and *Ppd* alleles have been found when deploying them in breeding [28]. For that reason, genome-wide association study (GWAS) has been adopted to confirm the loci associated with flowering time in durum wheat. A QTL associated with *Ppd-A1a* was shown to significantly reduce heading time in a recombinant inbred lines (RIL) population derived from the cross “Kofa” (“GS-100” allele) × “Svevo” (“GS-105” allele), suggesting that these two alleles have different effects on photoperiod sensitivity in durum [2]. A genome-wide association scan for heading date from 27 field trials in the Mediterranean Basin and in Mexico resulted in the identification of 50 chromosomal regions [29], of which only *Ppd-A1* and *Ppd-B1* had been previously described.

To better understand the effect of different flowering genes in durum wheat, a GWAS study was conducted using both modern germplasm and landraces and exposing these to very different climatic conditions, with the aim of triggering the expression of different loci and capturing their genetic effects. The use of a selected set of contrasting environments was deemed novel to better qualify and define the effect of different loci.

## 2. Results

### 2.1. Determination of Phenological Environments (PhEnv)

The plotting of climatic data (Figure 1) showed sizable variation among the 13 environments for average temperatures and day lengths, with Lebanon off-season planting in summer having a clear distinction in daylength and temperatures. Similarly, the experimental sites along the Senegal River (Fanaye and Kaedi) had a shorter daylength and higher temperatures. Terbol and Kfardan in Lebanon were the only sites where below-zero temperatures were recorded. A clustering analysis of climatic data based on PCA (Appendix A) revealed four main phenological environments (PhEnv) explaining 95% of the total climate variation and divided as follows: PhEnv1 included five environments in Morocco (TES16, MKZ15, MKZ16, MCH15, and MCH16), PhEnv2 represented three environments in Lebanon (TER15, TER16, and KFD16), PhEnv3 included two environments in both Senegal (FAN15 and FAN16) and Mauritania (KED15 and KED16), and PhEnv4 included only Terbol off-season in Lebanon (Figure 2).

### 2.2. Analysis of Variance for G, G x E, and G x PhEnv

Analysis of variance (Appendix A) showed significant (*p <* 0.01) differences for environment, PhEnv, genotype, genotype x environment, and genotype x PhEnv effects for all three tested traits. The GxPhEnv explained 68.7%, 80.7%, and 66.6% of the GxE for days to heading (DTH), cumulative growing degree days (CGDD), and cumulative day length (CDL), respectively. The DTH ranged from 71 to 107 days, with an average of 87 days across environments (Appendix A). The variation for CGDD and CDL ranged from 1049 to 2019 °C and 43,871 to 91,760 min, respectively (Appendix A). There was significant genotypic variation among accessions for DTH (Figure 3), with most of the modern lines flowering between 80 and 90 days, while landraces reached flowering only after 95–115 days. The high temperature and short photoperiod of PhEnv3 and PhEnv4 resulted in the shortest DTH (65 to 73 days), with the average min temperatures ranging between 14.4 and 21.6 °C and the average max temperatures from 31.0 to 35.5 °C. On the other hand, the short photoperiod and high temperatures of PhEnv3 prevented 3% of the landraces from ever reaching flowering, while PhEnv4 affected the possibility of flowering in 6% of the landraces and 2% of the modern lines.

To better define the effect of *Vrn* and *Ppd* genes at different PhEnv, Table 1 was developed to indicate how these loci will be differently expressed in different PhEnv and between them. PhEnv1 in Morocco experiences only mild cold temperatures, so would satisfy only weak *Vrn* requirements, but daylength increases in the transition to spring so that full *Ppd* requirements can be met. PhEnv2 in Lebanon has the same *Ppd* conditions as PhEnv1, but the colder winter temperatures favor flowering also in genotypes with stronger vernalization requirements. PhEnv3 along the Senegal River does not experience any cold temperatures, and its vicinity to the equator prevents significant changes in daylength during the season. Similarly, PhEnv4 does not experience cold days, and the photoperiod shortens between early summer planting and early fall harvest.

### 2.3. Marker-Trait Associations

Association analysis identified 41 and 68 markers associated with DTH, 34 and 63 markers for CGDD, and 27 and 66 markers for CDL for landraces and modern lines, respectively (Appendix A). Regression analysis confirmed 32 significant QTLs, 7 of which were in common between modern lines and landraces, 10 unique to landraces, and 13 unique to modern lines (Appendix A). Of these QTLs, six were significant for all the three traits, whereas four were unique for DTH and CDL and nine for both DTH and CGDD as well as for DTH alone in modern lines. For the landraces, eight QTLs were common among all traits, whereas five were unique for DTH and CDL, four for DTH and CGDD, and two for DTH. Based on these combinations, QTLs associated with DTH and CDL were defined as *ppd*, with DTH and CGDD as *vrn*, and with DTH alone and QTLs that overlap assigned as *eps*.

### 2.4. Flowering Loci Identified among Landraces

Among landraces, one QTL was identified only in PhEnv1 conditions, eight QTLs in PhEnv2 and PhEn4 conditions, and two QTLs by PheEnv3, while six QTLs showed significant effects across environments (Appendix A). Among the significant loci was also identified the marker tagging *Ppd-B1* (*Q.ICD.Ppd-05*) with a high LOD (4.6) r^2^ equal to 8.2% in PhEnv4 (Lebanon off season) and PhEnv2 (Lebanon main seasons). Moreover, *Q.ICD.Vrn-11* includes the marker tagging *Vrn-A1*, with a significant effect in PhEnv4 (Lebanon off season) explaining 12% of the variance. Similarly, three additional QTLs, *Q.ICD.Eps-01* on Chr1A, *Q.ICD.Ppd-10* on Chr4B, and *Q.ICD.Eps-14* on Chr6A, were significant in PhEnv4 and accounted for 12% to 30% of the total variance. *Q.ICD.Eps-03* and *Q.ICD.Ppd-04* on Chr2A and *Q.ICD.Ppd-05* on Chr2B showed significant effects across environments, explaining from 2.6% to 8% of the variance. Among them, *Q.ICD.Eps-03* was significant in the warm conditions in PhEnv4 (Lebanon off-season), explaining 12.8% of the total variance, whereas *Q.ICD.Ppd-04* was significant in PhEnv2 (Lebanon main season), where it accounted for 11.8% of the total variance. *Q.ICD.Ppd-02* (Chr1B) and *Q.ICD.Eps-07* (Chr3B) were significant in the warmest locations in PhEnv3 and PhEnv4, accounting for 7% to 15.5% of the variance. *Q.ICD.Eps-06* on Chr3A was significant in PhEnv1 (Moroccan stations) and PhEnv2 (Lebanon stations). In addition, *Q.ICD.Eps-08* and *Q.ICD.Eps-09*(Chr4A) and *Q.ICD.Vrn-12* (Chr5B), along with *Q.ICD.Vrn-16* and *Q.ICD.Eps-17* (Chr7A), showed significant effects in PhEnv2 (Lebanon), while *Q.ICD.Eps-13* (Chr6A) and *Q.ICD.Vrn-15* (Chr6B) were significant across environments and accounted for 4% to 14% of the variance for DTH.

### 2.5. Flowering Loci Identified among Modern Lines

Among modern lines, five QTLs influenced flowering in PhEnv1, 15 in PhEnv2, 12 in PheEnv3, and 10 in PhEn4, while 11 QTL in Across Env were significant (Appendix A). In particular, *Ppd-A1* (*Q.ICD.Ppd-19*) was consistently significant in all PhEnv. *Ppd-A1* showed the highest LOD (from 3.6 to 8.5), 0.17 r^2^, and explained 1.2% to 12.6% of the variation, while *Ppd-B* (*Q.ICD.Ppd-05*) was significant for PhEnv2 and across environments, accounting for 2.0% to 7.6% of the variance and 0.2 r^2^. Two vernalization-specific markers, *Vrn-A5* (*Q.ICD.Vrn-11*) and *Vrn-A7* (*Q.ICD.Vrn-16*), were significant for PhEnv3 (Senegal and Mauritania) and PhEnv4 (Lebanon off-season) only, and explained 0.03 r^2^ and 5.2% and 20% of the total variance, respectively. *Q.ICD.Ppd-19* was located in close proximity to *Ppd-A* with a high LOD (7.5), 0.13 r^2^, and significant variance (from 2.6% to 13.0%) in PhEnv3, PhEnv4, and across environments. Similarly, marker-AX-94956877 on Chr2B located adjacent to *Ppd-B* (thus, assigned common identifier *Q.ICD.Ppd-05*) explained 0.08 r^2^ and from 2.0% to 3.7% of the variance in PhEnv2 and across environments. *Q.ICD.Eps-20* included three different markers on Chr3A and was consistently significant in all PhEnv except PhEnv4, but explained 0.06 r^2^ and only 1.3% to 2.0% of the total variance. *Q.ICD.Eps-18* (Chr1B), 0.17 r^2^, and *Q.ICD.Eps-23* (Chr4B), 0.06 r^2^, were consistently significant in all pheno-environments except PhEnv1 and accounted for 2.5% to 24.0% of the variance. Additionally, *Q.ICD.Vrn-24* and *Q.ICD.Vrn-25* (located on Chr5A) and *Q.ICD.Vrn-26* on Chr5B were significant in PhEnv3 and accounted for up to 4% of the total variation. *Q.ICD.Eps-28* and *Q.ICD.Eps-29* on Chr6B, *Q.ICD.Eps-30* on Chr7A, and *Q.ICD.Eps-31-32* on Chr7B were significant in PhEnv2 and showed 2% to 7% of significant variation and 0.03 to 0.05 r^2^.

### 2.6. Common Loci between Landraces and Modern Lines

In addition to photoperiod- (*Ppd-B1*) and vernalization- (*Vrn-A5, VrnA7*) specific markers, landraces and modern germplasm shared four common loci for the control of flowering: *Q.ICD.Eps-03, Q.ICD.Eps-09*, *Q.ICD.Ppd-10,* and *Q.ICD.Vrn-15*, located on 2A, 4A, 4B, and 6B chromosomes, respectively (Table 2). However, these loci were seldom identified within the same PhEnv, and hence might have different allelic compositions between the two germplasm types. The results showed that *Ppd-B1* and *Q.ICD.Eps-09* were common to both groups of germplasm in PhEnv2, *Vrn-A1* in PhEnv3, *Q.ICD.Ppd-10* in PhEnv 4, and *Q.ICD.Vrn-15* across environments.

Several QTLs were dispersed among PhEnv and across environments. For instance, *Q.ICD.Eps-03* (Chr2A) was significant in PhEnv2 and PhEnv3 in modern lines, while in landraces it was significant in PhEnv4 and across environments. Similarly, *Q.ICD.Vrn-16* was significant in PhEnv2 in landraces, while in modern lines it was prominent in PhEnv3 and PhEnv4. In contrast, *Q.ICD.Eps-09, Q.ICD.Ppd-10*, and *Q.ICD.Vrn-11* were identified in the same PhEnv in both landraces and modern lines, with varying LOD (3.0 to 3.9), r^2^ (0.04 to 0.23) and variance (2.0% to 30.3%). Among these, *Q.ICD.Eps-09* was significant in PhEnv2 and *Q.ICD.Ppd-10* and *Q.ICD.Vrn-11* in PhEnv4. However, modern lines also showed significance in PhEnv3 for *Q.ICD.Vrn-11. Q.ICD.Vrn-15* was consistently significant across all locations and explained up to 0.04 r^2^ and 5.1% to 22% of the total variance in modern lines, while in landraces it was significant only in the combined analysis across environments and accounted for 0.18 r^2^ and 3.9% of the phenotypic variation.

### 2.7. Effect of Allelic Combinations

Four major QTLs (*Q.ICD.Eps-03, Q.ICD.Eps-09, Q.ICD.Ppd-10*, and *Q.ICD.Vrn-15*), two linked *Eps*, and one each from *Ppd* and *Vrn* were selected based on common position among germplasm, which reflected the maximum percent of variance and was used to define their additive effect in modern lines (Figure 4). The flowering effect of different allelic combinations at these loci was hence tested at the four PhEnv using the SNP call of the marker with the highest LOD underlying each QTL (Figure 4). ANOVA was performed to test these haplotype classes and the resulting LSD was used as a criterion to detect significant effect on the trait. Seven haplotypes were identified in the germplasm based on the four main markers tagging the QTLs. The germplasm harboring favorable alleles at *Q.ICD.Ppd-10* and *Q.ICD.Vrn-15* resulted in early flowering in all the PhEnv, irrespective of the allele at the other two *Eps* loci. However, the effects *of Q.ICD.Eps-03, Q.ICD.Eps-09* became prominent in PhEnv3 (Senegal and Mauritania) and PhEnv4 (Lebanon-off season), with genotypes revealing the earliest flowering. The haplotype class with the *Eps* gene revealed the minimum average flowering time of 48 and 52 days in Lebanon-off season and Senegal-Mauritania, respectively, whereas the maximum average flowering time was 129 days in Lebanon, followed by 110 days in Moroccan locations with the individual and combined effects of the *Ppd* and *Vrn* loci.

Apart from the common loci between the landraces and modern lines, few loci correspond to a specific flowering effect and indicated strong and weak effect for photoperiod and vernalization based on PhEnv. For example, *Q.ICD.Vrn-15* and *Q.ICD.Vrn-26* expressed in PhEnv1 belong to Morocco, where they showed a weak vernalization effect as they experienced a mild cold temperature, however *Q.ICD.Eps-06* in landraces and *Q.ICD.Eps-20* in modern lines on chr. 3A significantly affected the heading time in Moroccan stations (PhEnv1) and were neither involved in vernalization nor in photoperiod sensitivity, promoting their role as earliness per se (*Eps*). *Q.ICD.Vrn-12* on chr. 5B in landraces and *Q.ICD.Vrn-16* on chr. 7A in both germplasms showed a strong vernalization effect in Lebanon (PhEnv2), and *Q.ICD.Ppd-04* on chr. 2A had a strong photoperiod effect in PhEnv2. In modern lines, *Q.ICD.Vrn-24*, and *Q.ICD.Vrn-25* on chr. 5A indicated a very weak effect in Senegal and Mauritania as well as in Lebanon off-season due to the warm temperature. Similarly, these environments exhibited 12 h of continuous light, and lack of vernalization triggered the strong effect of earliness per se. Thus, *Q.ICD.Eps-07* on chr.3B in landraces and *Q.ICD.Eps-18* on chr. 1B in modern lines showed strong *Eps* effects. However, *Q.ICD.Ppd-02* on chr. 1B had a weak photoperiod response.

## 3. Discussion

### 3.1. Climatic Effect on the Control of Flowering Time

From an agricultural perspective, plants are considered better adapted to a specific region when they flower at the appropriate time [30]. The flowering time in durum wheat seems to involve more loci compared to bread wheat, with smaller effects, resulting in a higher diversity in flowering time, a driving force for adaptation to the diverse environments of the Mediterranean region [31]. A total of 384 durum entries assessed for flowering time across 13 environments located at different latitudes and temperature regimes (Figure 2 and Figure 3) confirmed a significant effect of all sources of variations for the three flowering traits considered: DTH, CGDD, and CDL. The accessions showed remarkable variation for these traits, with two main subgroups corresponding to landraces and modern lines. These results confirmed that temperature and photoperiod had a significant effect in determining the flowering initiation in addition to a number of genes promoting earliness per se (*Eps* genes). Four diverse pheno-environments (PhEnv) were determined based on the PCA of climatic conditions at the 13 environments. The four PhEnv had contrasting climatic conditions, and their pairwise comparison allowed us to distinguish the effects of *Ppd* and *Vrn* from those of the *Eps* loci (Table 1). Interestingly, the germplasm tested in PhEnv4 (warm summer of Lebanon) resulted in the shortest average DTH at 62 days, since only germplasm without the *Ppd* and *Vrn* requirement could flower. Similarly, PhEnv3 along the Senegal River also promoted early flowering time at 65–73 days, preventing germplasm with *Ppd* and *Vrn* requirements to flower. PhEnv1 of Morocco allowed for *Ppd* requirements to be met and mild *Vrn*, resulting in DTH ranging between 98 and 116 days. PhEnv2 represents Lebanese sites with long cold winter season, which highlighted the role of both *Ppd* and *Vrn* requirements, resulting in the longest DTH time from 122 to 137 days. Interestingly, CGDD also varied widely among PhEnv, with the longest season in number of days in Lebanon resulting in the lowest cumulative values (10,490–11,650), while the other PhEnv overlapped. CDL at flowering increased 5000–10,000 min from PhEnv3 to PhEnv4, while it was similar in PhEnv1 and PhEnv4. These results suggest that the main climatic effects have a strong role in determining the time to flowering. Past studies have shown that latitude integrates a number of variables affecting wheat development, among which the most important are photoperiod [32], temperature [33], and their interaction [34]. Similar studies in durum wheat [23] suggested photoperiod and temperature associated with different locations were the most important variables in explaining phenological differences. Another research in spring durum wheat indicated that photoperiod and temperature together explained 77% of the environmental variation between locations [35]. Our results provide only partial support to these findings, indicating that only up to 39% of phenotypic variation for flowering could be captured by PhEnv determined on the basis of climatic variables.

### 3.2. Known Loci Involved in the Control of Flowering Time in Durum Wheat

The present research confirmed a role for four well-established loci in controlling flowering time in durum wheat across the four PhEnv. Among them, three loci, *Ppd-B1*, *Vrn1*, and *Vrn3* on Chr. 5A and Chr. 7A showed significant effects in landraces as well as in modern germplasm, while *Ppd-A1* had a strong effect only in modern lines.

*Q.ICD.Vrn-11* included the *Vrn-A1* locus. It showed a high LOD value and accounted for significant variability (from 10% to 20%) in flowering time in modern lines and landraces in PhEnv3 (Senegal and Mauritania), and only in modern in PhEnv4 (Lebanon off-season), where no vernalization requirements can be met. This is in line with what can be expected, since accessions carrying the recessive *Vrn* allele would not be able to flower under those conditions. Previous studies suggested *Vrn-1* to be essential in the control of winter flowering types of common wheat [36,37,38], and durum wheat [23,28,39,40,41,42,43]. However, the results of the present research show only partial agreement with the literature, with a clear role of *Vrn1* confirmed only when no cold temperatures occur (see PhEnv3 and 4 in Table 1), but substantially no effect in PhEnv where weak or strong vernalization requirements can be met (see PhEnv1 and 2 in Table 1). Since PhEnv1 and 2 represent true Mediterranean environments where spring durum wheat is primarily cultivated, our results suggest the limited importance of *Vrn-A1* in determining the adaptation of durum wheat in its main area of production.

Earlier studies reported a significant role of *Vrn3* [5,44,45], suggesting its role in upregulating *Vrn-1* above the threshold levels required for flower initiation. *Q.ICD.Vrn-16* on chr. 7A herein reported likely corresponds to *Vrn3*. This QTL had a weak effect in modern germplasm in PhEnv3 and in landraces in PhEnv2. These results suggest a weak role for this locus in the control of flowering time in durum wheat, with no effect in environments where cold temperatures did not occur throughout the season (i.e., PhEnv4), probably due to the masking effect of *Vrn1*.

Locus *Q.ICD.Ppd-05* overlapped with *Ppd-B1* and showed a significant effect only in PhEnv2 (Lebanon), where full photoperiod requirements are presumably met for both modern lines and landraces. Past studies reported the importance of *Ppd-B1* in bread and durum wheat [23,29,31,46,47,48]. Royo et al. [23] reported the strong effect *Ppd-B1* in 35 spring durum wheat lines and an interaction between *Ppd-B1* and *Ppd-A1*. Recently, Würschum et al. [31] evaluated European durum genotypes for heading time in five environments and identified *Ppd-B1* as the major determinant of heading time in durum wheat, accounting for up to 26.2% of the variance. Our study observed confirmed an effect of *Ppd-B1*, but it accounted only for 5% of phenotypic variance in landraces as well as modern lines. This difference might be due to the wider genetic diversity assessed in our study as compared to research carried out previously.

Another major flowering gene reported by many researchers in durum and bread wheat is *Ppd-A1*. In the present study, *Q.ICD.Ppd-19* on chr. 2A spans the *Ppd-A1* locus. This was the most significant and stable QTL in our study, with effect in all PhEnv in modern germplasm, while it was monomorphic in landraces (no genetic variation available) as it can be expected. Earlier studies in durum wheat [49] also found a strong effect of *Ppd-A1* distributed in modern durum wheat lines, suggesting its exploitation started after the green revolution and was then further selected to increase adaptation. Wang et al. [50], Maccaferri et al. [51], and Royo et al. [23,28] also suggested a major role of *Ppd-A1* over *Ppd-B1* in durum wheat.

### 3.3. Identification of Novel Loci Involved in the Control of Flowering Time in Durum Wheat

The major loci known to be involved in the control of flowering time explained only partially the remarkable variation observed within the tested panel. Therefore, the association between the phenotypic and genotypic effects was investigated to identify a total of 28 loci not overlapping with known major flowering genes across the four PhEnv, including four loci in common between landraces and modern lines.

*Q.ICD.Eps-03* on chr. 2A was identified in PhEnv2 and PhEnv3 in modern germplasm for DTH and in PhEnv4 across environments in landraces for all the traits (DTH, CGDD, and CDL). Because its role was significant across contrasting environments, it is likely that its impact on flowering time is independent from climatic conditions (*Eps*), explaining up to 10% of the phenotypic variation. Giunta et al. [52] also reported a *QTL* on chr. 2A in durum wheat, explaining 16.9% of the phenotypic variation under long day, which falls within our QTL confidence interval (CI) when aligned to the Svevo genome [53].

*Q.ICD.Eps-09* was identified on chr. 4A and accounted for 2% to 18 % of the phenotypic variation in PhEnv2 (Lebanon) in both germplasm types. This PhEnv differs from all others because both the *Vrn* and *Ppd* requirements were presumably met. It is therefore likely that this locus is also acting as an earliness per se (*Eps*), with an effect visible only when the *Vrn* and *Ppd* requirements are fulfilled. Kamran et al. [54] also detected an overlapping QTL (*QFlt.dms-4A1*), which induced early flowering in spring wheat population.

*Q.ICD.Ppd-10* on chr. 4B controlled significant variation (3% to 30%) for both germplasm groups only in PhEnv4 (Lebanon off-season), and for modern lines only in the combined analysis across environments. PhEnv3 has similar climatic conditions to PhEnv4 (no *vrn* and no *ppd* requirements), but in PhEnv3 the photoperiod is decreasing rather than remaining constant throughout the season. Hence, this unique locus might be specifically linked to the promotion of earliness in durum germplasm cultivated at high latitudes during the summer cycle. Giunta et al. [52], Sanna et al. [55], and Milner et al. [56] reported QTLs linked to flowering time that overlap with *Q.ICD.Ppd-10* at 26.9 Mbp of chr. 4B and suggested it might correspond to the effect of the *Rht-B1* gene.

*Q.ICD.Vrn-15* chr. 6B was effective in modern germplasm in all environments except Lebanon (PhEnv2), while for landraces it was identified only in the combined analysis. PhEnv2 is the only environment for which complete vernalization requirements can be met. As such, this locus is likely involved in controlling vernalization requirements in durum wheat with a more refined mode of action compared to *Vrn1*. Giunta et al. [52] and Würschum et al. [31] also identified QTLs on chr. 6B at 590.8 Mbp with a significant effect on the determination of heading time. Hence, given its stronger sensitivity to temperature changes, it might be of value to consider this locus in addition to *Vrn1* to breed for earliness.

### 3.4. Define Usable Alleles for Earliness via Haplotype Analysis of Multiple QTLs

Four major QTLs (*Q.ICD.Eps-03, Q.ICD.Eps-09, Q.ICD.Ppd-10*, and *Q.ICD.Vrn-15*) were selected because they were identified in both germplasm types with some consistency across PhEnv. Haplotype analysis was conducted for these QTLs to define their additive nature across climatic conditions (Figure 4). In PhEnv1, the haplotype TATC resulted in significantly earlier modern lines than all other haplotypes, except CATC that matched. Comparison with the other haplotypes suggests a major role for the A allele of *Q.ICD.Eps-09* to promote earliness. In PhEnv2 again, TATC resulted in the earliest flowering haplotype, matching also CATC and CCTC and this time promoting the role of the TC combination for *Q.ICD.Ppd-10* and *Q.ICD.Vrn-15*. In PhEnv3, TATC was the earliest flowering type, matching four other haplotypes. Only the nucleotide T in *Q.ICD.Ppd-10* appears to be shared among all haplotypes, hence playing a pivotal role. Finally, in PhEnv4 the haplotype TATC resulted in significantly earlier flowering genotypes than all the other haplotypes, except for CATC. Comparison with the other haplotypes suggests a major role for the A allele of *Q.ICD.Eps-09* in promoting earliness. Considering these results together, the TATC haplotype appears to be the best combination to promote earliness across at all the tested conditions, and a good additive effect could be confirmed for at least three of the main QTLs (*Q.ICD.Eps-09, Q.ICD.Ppd-10,* and *Q.ICD.Vrn-15*), while it was not possible to discern the contribution of *Q.ICD.Eps-03*.

Across environments, two ICARDA’s elites, “Icavicre” and “Ouassara”, having the TATC haplotype, were the earliest flowering overall. In addition to the TATC haplotype, early flowering genotypes at individual PhEnv could be identified with different haplotypes, probably due to the additional effect of other loci. For instance, “CaMdoH25” and “Icambel” in PhEnv1 with the CCTC haplotype, “Bradano” and “Massara1” in PhEnv2 with CCTT and CATC, “IDON37-039” and “Moulsabil2” in PhEnv3 with CATC and CCTC, and “IDON37-053” and “Waha” in PhEnv4 with CCTC were the earliest flowering entries at individual sites.

## 4. Conclusions

The present study took advantage of contrasting climatic conditions to assess the effect of different loci on the control of flowering time. The testing of known major flowering loci explained only partially the large variation observed. The use of GWAS allowed us to define additional loci involved in the control of flowering for durum wheat, and the literature showed that other groups also identified the same. Hence, it appears that, for durum wheat, the flowering transition is controlled by only some of the loci known in common wheat, and that durum wheat breeders will have to rely also on the additional loci presented here to promote earliness. In particular, haplotype analysis revealed an important additive role for three of the identified QTLs, with a significant effect in controlling flowering time, besides the known major loci. The conversion of the markers underlying these loci in ready-to-use assays will promote their deployment by durum wheat breeders.

## 5. Materials and Methods

### 5.1. Plant Material

A durum wheat core collection comprised 96 landraces from 24 countries and 288 cultivars and elite breeding lines from eight countries. International Center for Agricultural Research in the Dry Areas (ICARDA) and International Maize and Wheat Improvement Center (CIMMYT) were used for this study. Detailed information regarding plant material is described in an earlier publication [57].

### 5.2. Phenotyping

A total of 13 field experiments were carried out in 2014–15 and 2015–16 in Morocco (Marchouche, Melk Zhar and Tassaout), Lebanon (Terbol in both main and off season and Kfardan), Senegal (Fanaye), and Mauritania (Kaedi). Out of 13 environments, two were rainfed and the remaining were irrigated. Full details of the environments are provided in Figure 2. The experiments were conducted according to an augmented design with 19 blocks and four repeated checks. Days to heading (DTH) was recorded as the number of days elapsed from the date of sowing to the onset of flowering determined at 50% of the plot with the tip of the spike emerging from the flag leaf (Zadoks scale stage 51). Daily records at each environment were minimum and maximum temperatures and the length of daylight in minutes. To estimate the cumulative growing degree days (CGDD) needed for flowering, the average daily temperature from planting to flowering were summed for each site following procedures described by Klepper et al. [58]. In case of wheat, a range of 0 to 32 °C temperature is considered optimal for growth; therefore, those values below and above these temperatures were converted to 0 and 32 °C, respectively. Cumulative day length (CDL) was instead measured as the total sum of minutes of sunlight needed from planting to heading at each environment. In the cases of Terbol off-season and four environments along the Senegal River, the high temperatures and short photoperiod prevented few modern lines and several landraces from flowering before harvest time. Nevertheless, these results were deemed of great interest for the analysis and so were converted to the maximum value for DTH, CDL, and CGDD recorded at each environment and used to run marker-trait association studies.

### 5.3. Genotyping

All the accessions were profiled by the 35K Affymetrix Axiom wheat breeders’ array (www.affymetrix.com) at Trait Genetics (Gatersleben, Germany) following the manufacturer’s instructions. In addition, three KASP assays were also run to characterize the variation for Ppd-A1, Ppd-B1, and *Vrn-A1* at LGC Genomic. Primer sequences and protocols for the KASP markers are available upon request to LGC clients. A total of 10 sub-populations were identified based on genetic diversity, as explained by Kabbaj et al. [57] and Sall et al. [59]. As described in an earlier publication [4,60], 7652 high-fidelity polymorphic single nucleotide polymorphism (SNPs) were obtained, showing less than 1% missing data, a minor allele frequency (MAF) higher than 5%, and a heterozygosity less than 5%. The sequences of these markers were aligned with a cutoff of 98% identity to the durum wheat reference genome [53] (available at: http://www.interomics.eu/durum-wheat-genome) to reveal their physical position. A sub-set of 500 highly polymorphic SNPs was selected on the basis of even spread along the genome and used to assess the population sub-structure, which revealed the existence of 10 main sub-groups [57]. To avoid discovery bias, these 500 markers were then removed from all downstream analyses. Linkage disequilibrium was calculated as squared allele frequency correlations (r^2^) in the TASSEL V 5.0 software [61] using the Mbp position of the markers along the bread wheat reference genome and plotted using the “Neanderthal” method. The linkage disequilibrium decay was measured at 51.3 Mb, as reported in Bassi et al. [60].

### 5.4. Statistical Analysis

The grouping of environments was conducted on the statistical software RX 64 (3.3.3) through stats package [62] via hierarchical clustering based on Euclidean distance calculated by principal component analysis (PCA), using as input the climatic variables measured in each environment: maximum temperature, average maximum temperature, minimum temperature, average minimum temperature, average cumulative growing degree days (CGDD), average days to heading (DTH), and average cumulative day length (CDL). The resulting clusters were defined as phenological environments (PhEnv). Combined analyses of variance were conducted across environments and PhEnv for DTH, CGDD, and CDL, assuming environment, PhEnv, and genotypes as fixed effects. Each year × location combination was considered as one environment. The best linear unbiased estimates (BLUEs) were derived for the individual environment, PhEnv, and across environments based on a linear mixed model. All the analyses were carried out with GENSTAT (version 2010) and the free statistical package RX64 version 3.3.3.

The broad-sense heritability was estimated as indicated by Falconer et al. [63]:H^2^ = [σ_g_^2^/σ_g_^2^] × 100,(1)
where σg^2^ is the genotypic variance and σp^2^ is the phenotypic variance. The genotypic and phenotypic variance components were estimated based on the method suggested by Burton and Devane [64]:σ_p_^2^ = σ_g_^2^ + σ_e_^2^+ σ_ge_^2^,(2)
σ_g_^2^ = [MS_g_ − MS_e_]/r,(3)
σ_ge_^2^ = [MS_ge_ − MS_e_]/r,(4)
where MS_g_ and MS_ge_ are the mean square due to genotype and GxE interaction, MS_e_ is the error mean square, and r is the number of replicates.

The association analysis was performed with TASSEL version 5.2.38. The marker-trait association test was carried out using mixed linear model (MLM) based on the kinship matrix estimated by Kabbaj et al. [57]. The analysis was performed using BLUE across environments and for each of the four PhEnv. In each case, the landraces and modern lines were analyzed separately. The significance of marker-trait association (MTA) was assessed by Boneforroni’s corrected equation. as suggested by Duggal et al. [65], assuming 288 marker trait hypothesis (12,000 Mbp of durum genome divided by LD decay of 51.3 Mb), which resulted in a LOD threshold of 3.0 and 3.4 for *p <* 0.05 and *p <* 0.01, respectively. Significant MTAs located at less than twice the LD decay distance were merged into one QTL. These QTLs were then further assessed by factorial regression to determine the true marker effect for each of the four PhEnv. Pearson’s critical value [66] for correlation was squared to obtain a critical r^2^ = 0.0225 for *p* < 0.05 and r^2^ = 0.0441 for *p* < 0.01, which were used as thresholds to determine significant effect on phenotypic variation.

To identify the best allelic combinations in different PhEnv, the four most significant QTLs were selected to define different haplotype classes of germplasm. These classes were then tested against the BLUEs for DTH at each PhEnv for the individuals within each class. A boxplot graph was constructed using the ggplot2 package [67] in RX64 version 3.3.3, and classes were tested for LSD differences assuming classes as fixed and genotypes as random factors.

## Figures and Tables

**Figure 1 plants-09-01628-f001:**
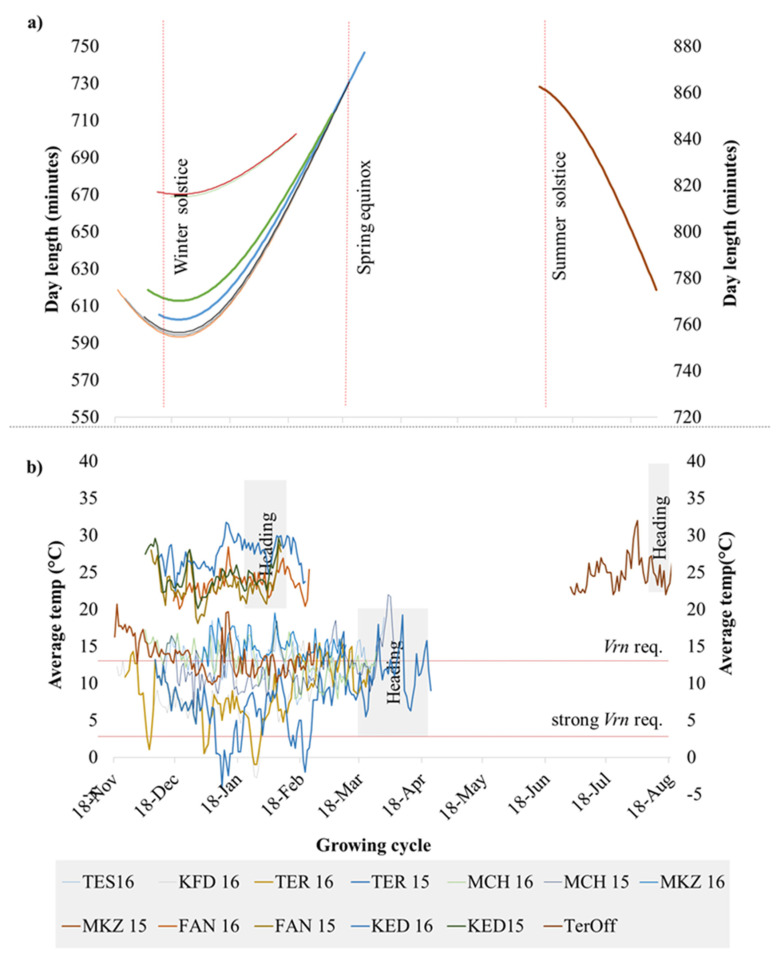
Distribution of (**a**) day length and (**b**) average temperature during the crop season at 13 environments including eight locations. The environments included are Kaedi (KED) 2015 and 2016 in Mauritania; Fanaye (FAN) 2015 and 2016 in Senegal; Marchouche (MCH) 2015 and 2016, Melk Zhar (MKZ) 2015 and 2016, and Tessaout (TES) 2016 in Morocco; Terbol (TER) 2015 and 2016, Kfardan (KFD) 2016, and Terbol off (TerOff) season 2016 in Lebanon. This last environment was plotted in the right axis to simplify visualization.

**Figure 2 plants-09-01628-f002:**
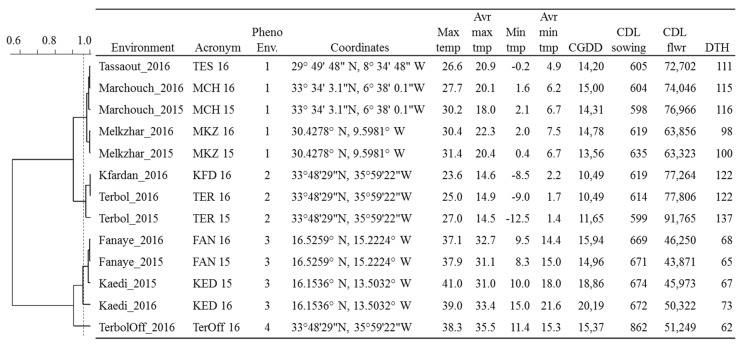
Meteorological parameters and phenological recordings for 13 environments including four countries—Morocco (Marchouche, Melk Zhar, and Tassaout), Lebanon (Terbol, Terbol off-season, and Kfardan), Senegal (Fanaye), and Mauritania (Kaedi)—used for the phenotyping of 384 durum wheat entries. The cladogram shows the relatedness of the environments based on PCA analysis of climatic data. Max temp: maximum temperature; Avr max temp: average maximum temperature; Min temp: minimum temperature; Avr Min Temp: average minimum temperature; CGDD: average cumulative growing degree days for flowering; CDL sowing: cumulative day length at the time of sowing; flwr: to flowering; DTH: average days to heading.

**Figure 3 plants-09-01628-f003:**
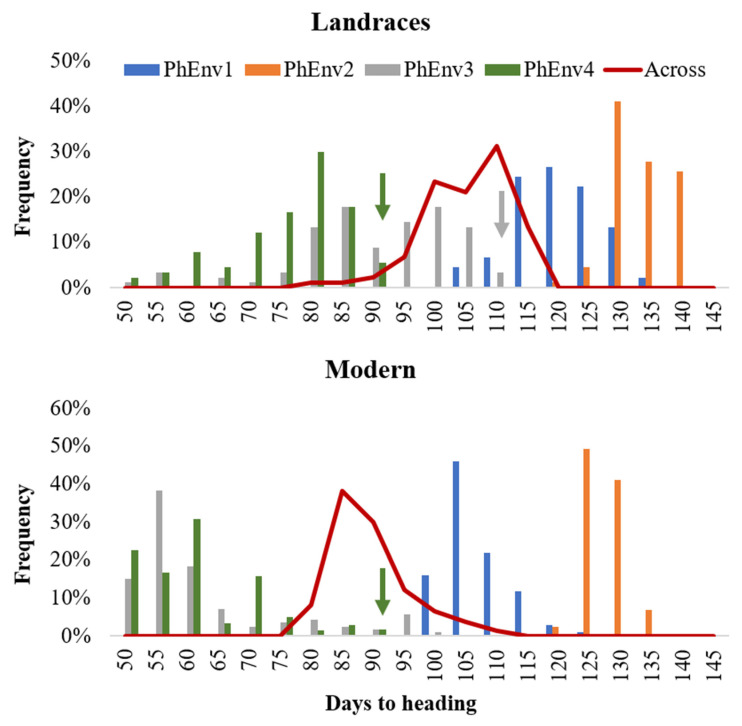
Distribution of 384 durum lines for days to heading in each pheno-environment (PhEnv) expressed as bars, combined across environments expressed as a red line, and divided for landraces and modern germplasm. The color-coded arrows indicate germplasm that did not flower in PhEnv3 or PhEnv4, to which was assigned the maximum value to be used for association studies.

**Figure 4 plants-09-01628-f004:**
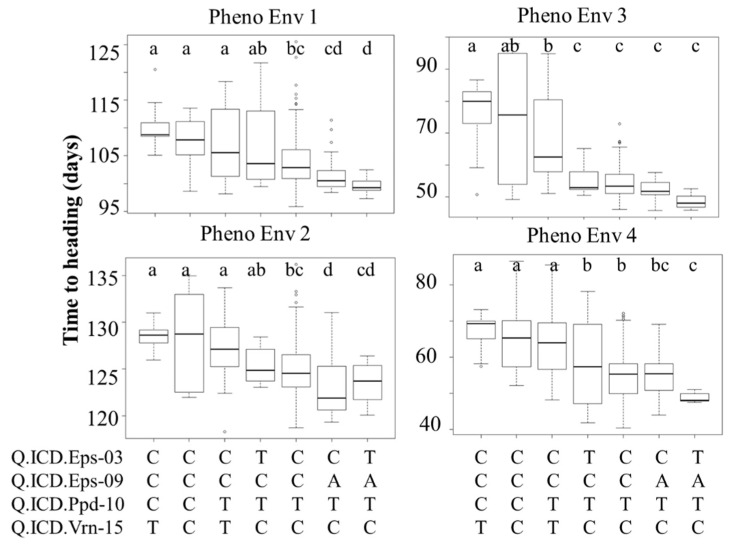
Boxplots showing days to heading (DTH) for four pheno-environments of durum core collection, grouped based on the haplotype score of four major QTLs found to be associated with the trait. The horizontal lines represent the average of each class, the box edges the 2nd and 3rd quartile, the whiskers the 1st and 4th quartiles, and the circles outliers. LSD classes are reported in the figure expressed as letters (a, b, c, d) to show significant variation between the haplotypes.

**Table 1 plants-09-01628-t001:** Matching of climatic requirements for flowering based on photoperiodism (*Ppd*) growing during the season to over 12 h of daylight, vernalization (*Vrn*) with extended periods of cold temperatures across four phenological environments (PhEnv), and pairing contrasts for their discrimination ability between genotypes.

PhEnv.	Location	Loci	Expected Effect	Differential Effects at Main Flowering Loci between Phenv		
				1	2	3
1	Morocco	*Ppd*	Yes			
		*Vrn*	Weak			
2	Lebanon	*Ppd*	Yes	Vrn: weak vs. strong		
		*Vrn*	Yes			
3	Senegal and Mauritania	*Ppd*	12h	Ppd: 12 h vs. normal Vrn: no vs. weak	Ppd: 12 h vs. normal Vrn: no vs. strong	
		*Vrn*	No			
4	Lebanon summer	*Ppd*	Shortening	Ppd: short vs. normal Vrn: no vs. weak	Ppd: short vs. normal Vrn: no vs. strong	Ppd: 12 h vs. shortening
		*Vrn*	No			

Here abbreviations Ppd, Vrn, and Eps used for photoperiod, vernalization, and earliness per se not for genes.

**Table 2 plants-09-01628-t002:** Significant QTLs and known flowering genes for landraces and modern lines across phenological environment (PhEnv). The numbers of circles explain the total percent of variance for days to heading explained by each QTL (● < 5%, ●● < 10%, ●●● < 20%, ●●●● > 20%).

QTL	Marker	Chr.	Germplasm	PhEnv 1	PhEnv 2	PhEnv 3	PhEnv 4	Across PhEnv
*Q.ICD.Ppd-05*	*Ppd-B1*	2B	Modern		●			●
			Landrace		●		●	
*Q.ICD.Vrn-11*	*Vrn-A1*	5A	Modern			●●●	●●	●
			Landrace			●●		
*Q.ICD.Vrn-16*	*Vrn-A3*	7A	Modern			●		
			Landrace		●			
*Q.ICD.Eps-03*	AX-94460586	2A	Modern		●	●		
			Landrace				●●	●
*Q.ICD.Eps-09*	AX-95630216	4A	Modern		●			
			Landrace		●●			
*Q.ICD.Ppd-10*	AX-94554200	4B	Modern				●●	●
			Landrace				●●●●	
*Q.ICD.Vrn-15*	AX-94711490	6B	Modern	●		●	●	●●
			Landrace					●

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
