# Peer review of "Genomic Regions Associated with the Control of Flowering Time in Durum Wheat"

_plants, 2020, doi:10.3390/plants9121628_

Round 1

Reviewer 1 Report

The manuscript showed QWAS for flowering time in durum wheat. Topic is interesting and fit for the aims and scope for “plants”. The experiments are well designed, and the work appears to have been carefully done. The results are clear. I think these finding will be of interest to readers and will accelerate the breeding of durum wheat. I have a few comments as below.

-L85: Detailed description about discrepancies should be mentioned here.

-Fig 1: Please add country names to the table.

-L128-129: Abbreviations (e.g. DTH, CGDD, etc.) should be full-spelled out since these words appear first time in the main text.

-L157-167: Some GWAS data should be presented in main text. These should not be presented as supplemental data because these are important data in this study.

-L346-380: Are there any candidate genes (e.g. PCL1) within the QTL region? Please mention about the candidate genes including PCL1.

-Some figs and tables are not cited in the main text.

Reviewer 2 Report

The manuscript is interesting and important for manipulating flowering time in durum wheat. Results were generally obtained with appropriate methods and well presented. Use of contrasting environments are attitude of this study.

Methods:

Method for KASP markers for Ppd-A1, Ppd-B1 and Vrn-A1 is not provided and not cited, also in line 61 – reference for functional SNP should be given.

Results:

Effects of some QTLs seems overestimated – Q.ICD.Vrn-28 is significant at p<0.05 and for very low r2 effect is 40% - this seems to be artefact, and normally at this kind of studies p<0.01 is more reliable. This phenomena should be better explained – is seems that groups compared may not be well represented and show high variation that excludes markers at higher stringency. Number of markers used for correcting of significance level is really low and should be better justified (is LD 51.3 Mb used as standard established with the genotyping method used in this paper?). It seems that authors should reanalyse QLTs with high effect at p<0.5 only. Can Q.ICD.Eps-03 be named major? (line 232, 383) – not better to exclude it as contribution was not discernible (398).  Lines 411-412 – it is not so apparent as durum is tetraploid so in common wheat more genes are ready to influence flowering time – this fragment seems too general and is not really supported by this research – for this analogous experiment should be performed at similar conditions and results analysed with the same assumptions.

Editorial remarks:

Generally, manuscript is well written. Comparisons with common wheat are often in manuscript (MS) so, unified nomenclature of Vrn genes should be used across MS and Vrn3 should be possibly replaced with Vrn-B3, and Vrn-A1 and Vrn-B1 consequently used (see lines 20, 48, 192, 198…).

Fig. 1. Headings is flwr defined flw (117)

131 strong … variation

182 18% should be 15.5%

190 11 were significant

Table S3. Is for modern cvs not landraces

Fig S1. Percent values for PCA1 and PCA2 should be supplemented

Table S2. QLTs Ppd-04/Ppd-05, Ppd-06/Ppd-07 are within 51.3 cM so should be merged or renamed.

Figure S2. Low quality – hard to read values at axes and to verify if ranges in text are fine.

Figure S3. Well presented

Reviewer 3 Report

Gupta et al. nicely present their study of the effects of different growing environments on a large GWAS panel of durum wheat, including both landraces and modern cultivars. To analyse environmental effects with more power they use a PCA approach to group their environments (PhEnv) which is an elegant approach to compare and contrast the major environmental inputs (vernalizing temperatures and photoperiod).

I only have minor suggestions to improve this manuscript:

  • For those of us with less knowledge of wheat, a line or so introducing the difference between durum and hexaploid wheat (e.g. genome composition), its particular uses and geographical regions of production would be appreciated. The genome composition in particular is relevant to the comparison between loci in hexaploidy vs. durum species.
  • What is ‘CDL sowing’?
  • Figure 2
    • The colours for Figure 2 are really hard to distinguish – some of this is to do with the image quality – could this be improved? Slightly thicker lines would also help.
    • The legend says that TerOff is plotted on the right hand axis – but that axis looks no different to the left one – is it the same? If so, I’d delete this note. It is clear that TerOff is plotted to the right because it was sown much later than the rest.
  • Figure S3 – although I appreciate the reasons for not showing loci with LOD of 0, it then becomes more difficult to get an idea of the proportion of loci that are linked and spacing of the markers. Ideally, I’d like to see a version of this figure with the 0 points included (perhaps in paler colours) but if not, could the authors include the percentage of 0 points in the legend? Possibly even with chromosomes marked?
  • In the results, the authors mention that some plants did not flower at all – in the methods it is mentioned that they were included in the GWAS analysis using the maximal values for the flowering within each site. This is fine – but could the authors include this information with the results section and indicate within Figure 3 in which bin these plants are included (if they are included in Fig. 3), or include a ‘did not flower’ bin within the histograms?

Very Minor suggestions

  • Line 83 – ‘of’, not ‘on’
  • 86 – GWAS doesn’t validate really – ‘confirm’ or ‘corroborate’ instead?
  • Line 95 – North African/Mediterranean environments don’t seem extremely different if you live in a cold, damp country! Perhaps ‘very’?!
  • Line 130 – 3 significant figures is plenty for the temperature and minutes. Decimal points.
  • Line 136 – prevented a few…from flowering at all.
  • Line 152 – pairing
  • Line 233 – common what?
  • Line 297 – ‘study’ for ‘research’

Author Response

Please find point-by-point response in the attachment
